**High-resolution (1 km) Polar WRF output for 79°N Glacier and the Northeast of Greenland**
**from 2014-2018**
Jenny V. Turton[1], Thomas Mölg[1], Emily Collier[1]
[1]Climate System Research Group, Institute of Geography, Friedrich-Alexander University, Erlangen-
Nürnberg, 90158, Germany.
*Correspondence to:* Jenny V. Turton (jenny.turton@fau.de)
**Abstract**
**The northeast region of Greenland is of growing interest due to changes taking place on the**
**large marine-terminating glaciers which drain the north east Greenland ice stream.**
**Nioghalvfjerdsfjorden, or 79°N Glacier, is one of these that is currently experiencing**
**accelerated thinning, retreat and enhanced surface melt. Understanding both the influence of**
**atmospheric processes on the glacier and feedbacks from changing surface conditions is crucial**
**for our understanding of present stability and future change. However, relatively few studies**
**have focused on the atmospheric processes in this region, and even fewer have used high-**
**resolution modelling as a tool to address these research questions. Here we present a high**
**spatial- (1 km) and temporal- (up to hourly) resolution atmospheric modelling dataset,**
**NEGIS_WRF, for the 79°N and northeast Greenland region from 2014-2018, and an evaluation**
**of the model's success at representing daily near-surface meteorology when compared with**
**automatic weather station records. The dataset, (Turton et al, 2019b:**
**doi.org/10.17605/OSF.IO/53E6Z), is now available for a wide variety of applications in the**
**atmospheric, hydrological and oceanic sciences in the study region.**

**1.  Introduction**
The surface mass balance of a glacier is largely controlled by regional climate through varying mass
gains and losses in the ablation and accumulation zones, respectively. The large amount of mass lost
from the Greenland Ice Sheet (GrIS) within the last few decades (approximately 3800 billion tonnes of
ice between 1992 and 2018: Shepherd et al., 2020) has largely been located around the coast of
Greenland, due to the thinning and retreat of marine-terminating glaciers (Howat & Eddy, 2011), and
the surface mass loss in the ablation zone due to enhanced melting and runoff (Rignot, et al., 2015;
van den Broeke et al., 2017). A recent study found that enhanced meltwater run off, connected to
changing atmospheric conditions, was the largest contributor of mass loss for Greenland (52%)
(Shepherd et al., 2020). The remaining 48% of mass loss (1.8 billion tonnes of ice) was due to
enhanced glacier discharge, which has been increasing over time (Shepherd et al., 2020).

The majority of studies of the surface mass loss in Greenland and its atmospheric controls are largely constrained to southern and western Greenland (e.g Kuipers Munneke et al., 2018; Mernild et al., 2018), or to specific warm events such as the 2012 melt event (e.g Bennartz et al., 2013; Tedesco et al., 2013). However, recent studies have shown that the northeast of Greenland, specifically the North East Greenland Ice Steam (NEGIS) is now experiencing high ice velocity and accelerated thinning rates (Joughin et al., 2010; Khan et al., 2014). NEGIS extends into the interior of the Greenland ice stream by 600 km and three marine-terminating glaciers connect the NEGIS with the ocean. The largest of these glaciers is Nioghalvfjerdsfjorden, often named 79°N after its latitudinal position. Until recently, very few studies focused on 79°N glacier and NEGIS as they were thought to contribute little to surface mass loss and instabilities (Khan et al., 2014; Mayer et al., 2018). However, 79°N glacier, with its 80 km long by 20 km wide floating tongue, has retreated by 2-3 km between 2009 and 2012, and the surface of the tongue and part of the grounded section of the glacier are now thinning at a rate of 1 m yr$^{-1}$ (Khan et al., 2014, Mayer et al. 2018). The glacier is at a crucial interface between a warming ocean and a changing atmosphere. The mass loss from the floating tongue is largely attributed to basal melting due to the presence of warm (1°C) ocean water in the cavity below the glacier (Wilson & Straneo, 2015, Schaffer et al., 2017, Münchow et al., 2020). However, even the grounded part of the glacier is characterised by large melt ponds and drainage systems (Hochreuther, P. pers. comm); suggesting that atmospheric processes may also be at play. Furthermore, atmospheric processes may be responsible for driving the warm Atlantic water under the glacier tongue, which leads to melting of the glacier base (Münchow et al., 2020). 79°N glacier is of further interest because its southerly neighbour, Zachariae Istrom, recently lost its floating tongue (Mouginot et al., 2015).

A number of studies have used atmospheric modelling as a tool to investigate the region, although they have largely been confined to short case studies (Turton et al., 2019a), focused on past climates (e.g 45000 years ago by Larsen et al., 2018), or targeted specific atmospheric processes (Leeson, et al., 2018; Turton et al., 2019a). There are a number of atmospheric models that have been applied to the Greenland region, however these are often run at a resolution that is too coarse to resolve the 79°N glacier, especially its floating tongue, which can therefore be missing in many simulations. These data are usually statistically downscaled to calculate the surface mass balance of the glacier, using a digital elevation model and a shape file of the glacier. The resolution of the atmospheric models used in published studies for Greenland generally exceed 10km: e.g the Modèle Atmosphérique Régional (MAR) at 20-km (Fettweis et al, 2017) RACMO2 at 11-km (Noël et al., 2016) and HIRHAM5 at 25-km (Mottram et al., 2017a). Recently, there have been attempts at modelling the polar regions using non-hydrostatic regional climate models, including HARMONIE-AROME at 2 km resolution for the Southwest of Greenland (Mottram et al., 2017b), and the NHM-SMAP at 5 km resolution for the whole of Greenland (Niwano et al., 2018). However, the Mottram et

al. (2017b) study does not include the northeast of Greenland. Furthermore, the focus of the Niwano et al. (2018) study was to improve the surface mass balance estimates, as opposed to providing output for a more general atmospheric sense, and the model was not convection permitting. In convection-permitting models, typically for spatial resolutions higher than 5km, convection begins to be explicitly resolved. This can enhance the representation of convection and associated precipitation, as opposed to using a convection parameterisation scheme, (Pal et al., 2019). As yet, there are no very high-resolution, multi-year atmospheric datasets available for the northeast of Greenland or the wider region.

Here, we address this data gap by presenting a 5-year (2014-2018), high-resolution (1 km) atmospheric simulation using a polar-optimised atmospheric model and evaluate its skill in representing local meteorological conditions over the 79°N region in northeast Greenland. The dataset is named NEGIS_WRF after its location of focus and model used. As the 79°N region is of growing interest, this data could be beneficial for numerous other studies and applications. Indeed, current ongoing research as part of the Greenland Ice sheet-Ocean interaction (GROCE) project (www.groce.de, last accessed October 1 2019) include using this data for surface mass balance studies and to investigate the relationship between specific atmospheric processes and surface melt patterns. For studies of the surface mass balance of the NEGIS, further downscaling would not be necessary. With a horizontal resolution of less than 5km, many atmospheric processes are accurately resolved including katabatic winds and warm-air advection (Turton et al., 2019a). Furthermore, high-resolution output is crucial for the complex topography on the northeast coast, where steep and variable topography can channel or block the winds, and lead to strong variability of the radiation budget. The WRF dataset is also intended as input to an ocean model, used in an ocean-glacier interaction study, input into a hydrologic model and for an ice sheet modelling study. Here we present an evaluation of the ability of NEGIS_WRF at representing key near-surface meteorological and radiative conditions, to demonstrate the applicability of the dataset for these and other studies in the atmospheric, cryospheric and oceanic fields.

**2. Data and Methods**

**2.1 Model Configuration**

The Polar Weather Research and Forecasting (Polar WRF) model is a version of the WRF model that was developed and optimised for use in polar climates (Hines et al., 2011). The non-hydrostatic WRF model (available online from http://www.mmm.ucar.edu/weather-research-and-forecasting-model; last accessed July 29 2019) has been widely used for both operational studies and for research in many regions, and at many scales (Powers et al., 2017; Skamarock & Klemp, 2008). The current version of polar WRF used here is v3.9.1.1, which was released in January 2018, and is available from http://polarmet.osu.edu/PWRF/ (last accessed July 29 2019). Polar WRF has been developed for use in the Arctic and Antarctic by largely optimising the Noah Land Surface Model

(LSM) (Chen & Dudhia, 2001) to improve heat transfer processes through snow and permanent ice,
and by providing additional methods for sea-ice treatment (Hines et al, 2015). For a full description of
the Polar WRF additions, see (Hines & Bromwich, 2008; Hines et al., 2011; Hines et al., 2015) and
citations therein.

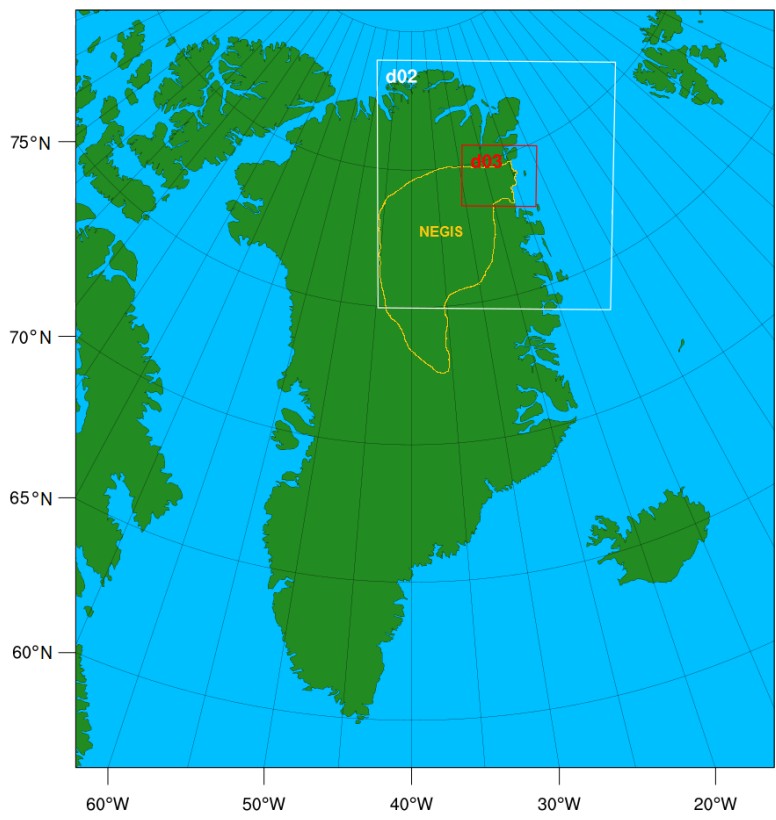

**Figure 1: The domain configuration for the Polar WRF runs and the approximate outline of**
**NEGIS following Khan et al. (2014).**

The meteorological initialisation and boundary input data is from the ECMWF (European

Centre for Medium range Weather Forecast) ERA-Interim dataset at 6-hourly intervals (Dee et al.,
2011). This reanalysis product was more accurate at resolving mesoscale processes in the northeast of
Greenland compared to MERRA2 reanalysis data and has previously been used for Polar WRF
simulations in Greenland (DuVivier & Cassano, 2013; Turton et al., 2019a). The Sea Surface
Temperature (SST) and sea ice concentration values are from the NOAA Optimum Interpolation
0.25° resolution daily data. This is a combination of data from the Advanced Very High Resolution
Radiometer (AVHRR) infrared satellite and Advanced Microwave Scanning Radiometer (AMSR)
(doi:10.5065/EMOT-1D34, data retrieved from https://rda.ucar.edu/datasets/ds277.7/, last accessed
July 29 2019). In-situ ship and buoy data are used to correct satellite biases, leading to relatively low
mean biases of 0.2-0.4K for SST data (more information on this dataset can be found in Banzon et al.,
2016). This higher resolution dataset was required due to the very blocky coastline in the SST and sea
ice data from ERA-Interim. The domain setup is shown in Figure 1. The outermost domain (D01) is at
25km, D02 is 5km and D03 (innermost) is 1km grid spacing. Boundary conditions, including sea ice
fraction and SST were updated every 6-hours. Analysis nudging was used in the outer domain (D01)
to constrain the large-scale circulation while allowing the model to freely simulate in D02 and D03.
Nudging is the process of constraining the interior of model domains towards the larger-scale field
(from reanalysis data) which drive the simulation (Lo et al., 2008., Otte et al., 2012). It has been
found to improve simulations of the large-scale circulation (Bowden et al., 2012) and reduce errors in
the mean and extreme values (Otte et al., 2012) from relatively long runs. We only nudge the outer
domain (D01) to allow the higher-resolution domain to evolve freely. The USGS 24 category landuse
and landmask was adjusted using the European Space Agency (ESA) Climate Change Initiative (CCI)
landuse product, to provide a better representation of the glacier outlines and the terminus of the
floating tongue (https://www.esa-landcover-cci.org/, last accessed September 5 2019). A number of
open-water grid points were manually changed to glacierised during January-June and September-
December to better represent the floating tongue of the Spalte Glacier (tributary of 79°N on the
northeast side) and the sea ice in the adjacent Dijmphna Sound (Fig. 2). Other small exposed water
areas along the coast, which are permanently frozen except in July and August each year
(Hochreuther, P., 2019 personal communication), were also changed to ice during all months except
July and August (Fig. 2). The glacier extents are treated as static throughout the run, which is an
appropriate approximation given the small and likely negligible area of calving of 79°N during our
study period (see ENVEO, 2019 for calving front locations from 1990 to 2017). There are 60 levels in
the vertical, with a 10-hPa model top and a lowest model level ~16m above the surface.
Many of the parameterisations for the model configuration were selected based on numerous,
previous Polar WRF runs over Greenland and the Arctic (for example Hines et al., 2011). In brief, the
following parameterisations were employed: the Noah LSM (Chen & Dudhia, 2011), due to its
optimisations that have been tested over Greenland (Hines & Bromwich, 2008), Arctic sea ice (Hines,
et al 2015) and Arctic land (Hines et al., 2011); the Morrison two-moment scheme for microphysics,
which has been shown to out-perform other schemes in both Polar regions (Bromwich, et al., 2009;
Lachlan-Cope, et al., 2016; Listowski & Lachlan-Cope, 2017); the Eta Similarity Scheme for surface
layer physics (Janjić, 1994) and the Yonsei University Scheme for planetary boundary layer
parameterisation. This was used due to the topographic wind scheme (Hong et al., 2006) that can
correct excessive wind speeds in areas of complex topography, such as the northeast coast of
Greenland (employed in D02 and D03 only, where complex orography is best resolved). Further
parameterisations include: the Kain-Fritsch scheme for cumulus convection (Kain, 2004) (D01 and
D02 only, as the resolution of D03 allows convection to be explicitly resolved); and, the Rapid
Radiative Transfer Model (RRTM) longwave and Goddard shortwave schemes for radiation, based
on sensitivity testing for the polar regions by Hines et al. (2008) and subsequent runs over Greenland
(DuVivier & Cassano, 2013; Hines et al., 2011). Whilst the majority of these options were selected
for testing based on the works of other publications, a short sensitivity study was also conducted,
alongside with testing the horizontal and vertical resolution and locations of the domains (not
included). It was found that a combination of the options above were best suited to the northeast of
Greenland when compared with observations on the floating tongue of the 79°N glacier from 1996-
1999 (Turton et al., 2019a).

Other options specified for this study include using a fractional sea ice treatment, which
allows calculation of different surface temperature, surface roughness and turbulent fluxes for open
water and sea ice conditions within the grid cell, and then calculates an area-weighted average for the
grid (DuVivier & Cassano, 2013; Hines et al., 2011). The adaptive timestep was used to optimise the
simulation speed. For each year simulated, the model was initialised on September 1 before the onset of
the accumulation season and ran continuously until October 1 of the following year (e.g September 1
2016 - October 1 2017). September was then discarded as a spin up month. The model produces
similar magnitude snow depths to available observations (Pedersen et al. 2016). Due to limited
snowfall and snow depth observations in this region, we compared cumulative snowfall to ERA5
products during testing, which have been shown to have a relatively good agreement with
observations by Wang et al. (2019). The maximum snow depth and average annual accumulation
were well captured by Polar WRF compared to ERA5.

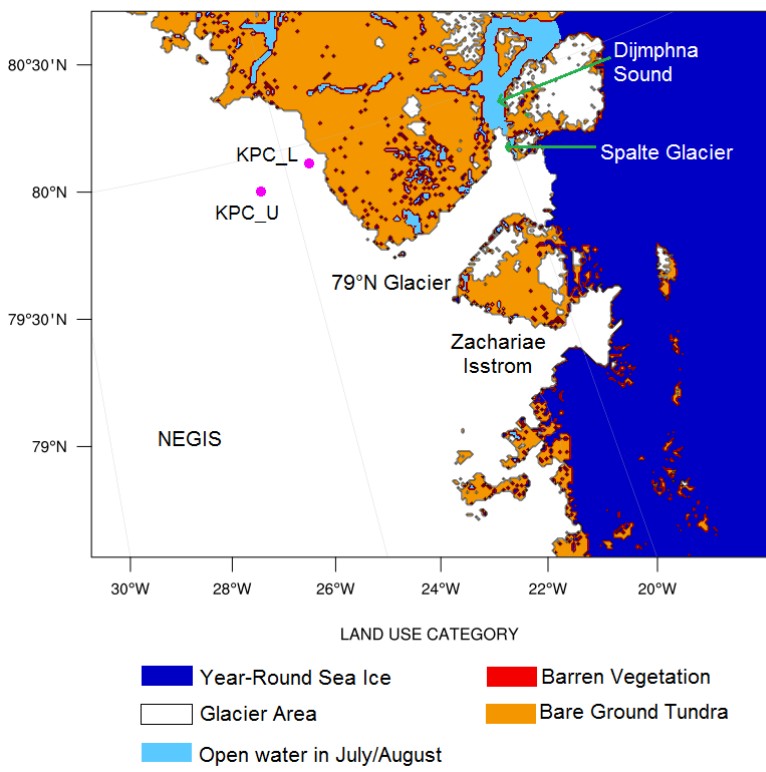


**185 Figure 2: A map of the land use types for D03. Colours represent the land use type, except for**
**186 light blue, which highlights the manually changed land use from open water to sea ice during**

**winter. Important locations are also highlighted, as are the locations of the two AWS sites (pink**
**dots).**

The data were output at hourly intervals for D03, at six-hourly intervals for D02 and at daily intervals
for D01. Daily mean values for key meteorological variables from D02 and D03 were calculated from
the hourly values and are available along with the daily instantaneous values from D01 at the Open
Science Framework repository (Turton et al. 2019b: doi.org/10.17605/OSF.IO/53E6Z).

**2.2 Observational Data**
The remote nature of the location of interest provides few in-situ observational datasets for model
evaluation. However, the PROMICE (Programme for Monitoring of the Greenland Ice Sheet) network
(www.promice.dk, last accessed October 1 2019; van As & Fausto, 2011), operated by the Geological
Survey of Denmark and Greenland (GEUS) has two permanent Automatic Weather Stations (AWSs)
available for comparison of daily means of meteorological variables and a number of surface energy
balance components. The AWSs are referred to as KPC_L and KPC_U due to their location on
Kronprincs Christian Land (located to the northwest of 79°N glacier; see Table 1 for AWS details of
location, dates and available variables. Although hourly data are available, daily means are used for
evaluation due to the multi-year timescale of the study, but the authors note that an evaluation of
hourly data should be performed before using these data for analysis at these time scales. Please refer
to van As & Fausto, (2011) and Turton et al., (2019a) for more information on the PROMICE data in
this location (doi.org/10.22008/promice/data/aws, available at www.promice.dk, last accessed
October 1 2019). Observations are not taken at exactly 2m above the surface but vary with
accumulation and ablation. Over bare ice, the sensor is 2.6m above the surface (van As et al., 2011).
To clarify that the observations represent near-surface conditions, and are compared with 2m and 10m
model output, we use the abbreviation X2 or X10 to represent both modelled and observed variables
at the respective heights. The mean values from the observational data are calculated from daily
averages from January 1 2014- December 31 2018 to keep a consistent period across all data.
The in-situ AWS observational data are used to evaluate the NEGIS_WRF output and to
provide a judgement of its skill to benefit future users. The focus of the evaluation is to test WRF's
ability to represent local meteorological conditions over a polar glacier. Daily mean values from
NEGIS_WRF have been calculated from hourly output at the location of the two AWSs. All
evaluation focuses on near-surface meteorological output from D03.

**Table 1: The location, elevation and data availability of the two AWSs used for model**
**evaluation. We evaluate the model output with four variables from the AWSs. Data was**
**unavailable at KPC_L between January 15 2010 and July 17 2012 due to retrieval problems. T**
**is air temperature, Q is specific humidity, WS and WD are wind speed and direction,**

**respectively. Observations are taken at approximately 2m above the surface, but this does vary with accumulation and ablation (see section 2.2). Sensor error estimates come from the sensor manufacturers. See van As & Fausto (2011) for more information on sensors and observations**.

| Name | Location | Elevation (m a.s.l) | Data Availability | Variables used for evaluation | Sensor Error Estimates |
|---|---|---|---|---|---|
| KPC_L | 79.91°N, 24.08°W | 380 | 01.01.2009- present | T, Q, WS, WD, SW$_{down}$, LW$_{down}$ | T: ± 0.2°C RH: ± 1.5% WS: ± 0.3ms$^{-1}$ WD: ± 3° Radiation: 10% |
| KPC_U | 79.83°N, 25.17°W | 870 | 01.01.2009- 14.01.2010, 18.07.2012-present | T, Q, WS, WD, SW$_{down}$, LW$_{down}$ | T: ± 0.2°C RH: ± 1.5% WS: ± 0.3ms$^{-1}$ WD: ± 3° Radiation: 10% |

## 3. Results

### 3.1 Model evaluation: Daily Means

The air temperature is simulated well by the WRF simulations with a coefficient of determination ($R^2$) of 0.92 at both KPC_L and KPC_U (Table 2, Fig 3). Similarly, the mean biases and RMSE are small. The mean bias and RMSE are slightly larger during winter (DJF) at KPC_U, but overall, the $R^2$ value at both locations remains above 0.64. The particularly low daily temperatures observed during winter at KPC_U are not fully captured by the WRF simulations (Fig. 3b). The model can, however, capture the larger variability in winter (Fig. 3), including 'warm-air events', where the air temperature increases by more than 10°C in a few days, leading to temperatures above the average for winter (Turton et al., 2019a). Figure 4 presents the near-surface air temperature and 10m wind vectors for June 6 2015, to show what the temperature and wind fields look like for an example time period during the ablation period (June to August). The onset of the ablation season is earlier over the floating tongue of the glacier, as seen by the above freezing air temperatures at low elevations in Figure 4. WRF simulates the humidity very well annually and during winter for both locations. The humidity during summer is slightly less well simulated, with mean biases of 0.4 and 0.6 g/kg for KPC_L and KPC_U respectively (Table 2). However, the $R^2$ values remain above 0.44 for the summer season. For both locations, annually and seasonally, WRF is moister than in observations, however the mean biases remain relatively small (less than 0.6 g/kg), and the differences are not

statistically significant except for during summer at KPC_U (which is statistically different at the 99%
confidence level using a student t-test). The wind direction in WRF deviates more from the AWS data
than for temperature and moisture, which is likely due to the particularly steep and complex
topography of the region which may not be accurately represented by the model, even at 1 km
resolution. The largest bias is an annual bias at KPC_L (10.7°) as WRF simulates the wind direction
predominantly more northerly than in observations (Table 2), which leads to poor $R^2$ values (0.01) and
high RMSE. For KPC_U annually and seasonally, the biases remain at or below 8.6° and $R^2$ values
are 0.36, which shows that WRF is capable of representing the wind direction at KPC_U. Some of
these errors may relate to measurement errors of the wind senor, which is ±3° (see Table 1). The
model performs better at simulating the wind speed than the wind direction. Annually and during
winter, the $R^2$ values are relatively high (above 0.31) at both locations, and mean biases remain at or
below 2.3 ms$^{-1}$ both annually and seasonally. None of the biases between WRF and observations are
statistically significantly different for daily mean wind speed or air temperature (Table 2).

Shortwave and longwave radiation values are important for a range of possible future studies

including input to surface mass balance and ocean models. Therefore, we have validated the
NEGIS_WRF output for both the downwelling shortwave and longwave by comparing it to
observations at the two sites (Table 2). Annually, the biases are within sensor error range (Table 1)
and differences between WRF and observations are not statistically significant for both downwelling
shortwave ($SW_{down}$) and longwave ($LW_{down}$). Due to the lack of sunlight during winter at this latitude,
the $SW_{down}$ biases and RMSE are small and the $R^2$ values (0.78 and 0.75 for KPC_L and KPC_U
respectively) are high for both locations (Table 2).  The mean biases are largest for $SW_{down}$ during
summer, but a relatively high $R^2$ value shows that WRF still has a great deal of skill (0.82 at KPC_U).
Biases for $LW_{down}$ are largest during winter (-10.3 and -15.3 Wm$^{-2}$ at KPC_L and KPC_U
respectively), which is likely a product of increased wintertime variability due to storm frequency and
location (van As et al., 2009). Similarly, Cho et al. (2020) found that biases of $LW_{down}$ compared to
satellite observations were larger for the Morrison microphysics scheme (which we use here) than for
the WRF single moment 6-class scheme. However, it was concluded that Polar WRF has the ability to
accurately simulate the spatial distribution of Arctic clouds and their optical properties with both
tested schemes (Cho et al., 2020). None of the differences between WRF output and observations for
the radiation components were statistically significant (Table 2).

**Table 2: Comparison of the near-surface WRF model output to AWS data at KPC_L and**
**KPC_U. ANN refers to annual mean values, DJF refers to winter average values whereas JJA**
**refers to summer average values. * refers to statistically significant differences between WRF**
**and AWS at the 99% confidence interval, using the student's t-test.**

| Variable (units) | Location | AWS Mean | Mean Bias | RMSE | $R^2$ |
|---|---|---|---|---|---|

| | | | (WRF-AWS) | | |
|---|---|---|---|---|---|
| T2 ANN (°C) | KPC_L | -13.6 | -0.3 | 3.0 | 0.92 |
| | KPC_U | -17.2 | 1.8 | 4.0 | 0.92 |
| T2 DJF (°C) | KPC_L | -23.3 | 0.0 | 3.2 | 0.86 |
| | KPC_U | -27.6 | 2.6 | 5.2 | 0.64 |
| T2 JJA (°C) | KPC_L | 1.6 | -1.8 | 2.6 | 0.71 |
| | KPC_U | -1.5 | -0.1 | 1.9 | 0.69 |
| Q2 ANN (g/kg) | KPC_L | 1.6 | 0.2 | 0.4 | 0.92 |
| | KPC_U | 1.4 | 0.3 | 0.5 | 0.92 |
| Q2 DJF (g/kg) | KPC_L | 0.4 | 0.1 | 0.1 | 0.81 |
| | KPC_U | 0.3 | 0.1 | 0.2 | 0.66 |
| Q2 JJA (g/kg) | KPC_L | 3.2 | 0.4 | 0.8 | 0.44 |
| | KPC_U | 3.0 | 0.6* | 0.9 | 0.56 |
| WD10 ANN (°) | KPC_L | 219.4 | 10.7* | 74.3 | 0.01 |
| | KPC_U | 277.9 | 3.4 | 29.9 | 0.36 |
| WD10 DJF (°) | KPC_L | 238.5 | -3.2 | 49.9 | 0.01 |
| | KPC_U | 274 | 8.6 | 29.1 | 0.36 |
| WD10 JJA (°) | KPC_L | 211.6 | 6.8* | 80.2 | 0.01 |
| | KPC_U | 279.9 | -0.1 | 31.7 | 0.25 |
| WS10 ANN (m/s) | KPC_L | 5.7 | 0.4 | 2.9 | 0.42 |
| | KPC'_U | 4.8 | 1.5 | 2.5 | 0.49 |
| WS10 DJF (m/s) | KPC_L | 6.4 | 1.0 | 3.2 | 0.50 |
| | KPC_U | 5.2 | 2.3 | 3.4 | 0.38 |
| WS10 JJA (m/s) | KPC_L | 5.4 | -0.8 | 2.7 | 0.31 |
| | KPC_U | 4.2 | 0.8 | 1.9 | 0.45 |
| $SW_{down}$ ANN ($Wm^{-2}$) | KPC_L | 114.5 | 4.7 | 34.1 | 0.94 |
| | KPC_U | 124.6 | 3.8 | 23.8 | 0.97 |
| $SW_{down}$ DJF ($Wm^{-2}$) | KPC_L | 0.1 | -0.1 | 0.4 | 0.78 |
| | KPC_U | 0.2 | -0.1 | 0.5 | 0.75 |
| $SW_{down}$ JJA ($Wm^{-2}$) | KPC_L | 271.6 | 13.1 | 62.3 | 0.63 |
| | KPC_U | 295.1 | 11.9 | 42.2 | 0.82 |
| $LW_{down}$ ANN ($Wm^{-2}$) | KPC_L | 212.0 | -7.1 | 24.7 | 0.76 |
| | KPC_U | 202.5 | -9.2 | 26.1 | 0.71 |
| $LW_{down}$ DJF ($Wm^{-2}$) | KPC_L | 181.9 | -10.3 | 26.8 | 0.50 |
| | KPC_U | 179.6 | -15.3 | 31.6 | 0.40 |

| LW$_{down}$ JJA (Wm$^{-2}$) | KPC_L | 267.3 | -4.9 | 23.8 | 0.38 |
|---|---|---|---|---|---|
| | KPC_U | 250.8 | -6.4 | 21-6 | 0.49 |


The larger RMSE and lower R$^2$ values during summer for wind direction can, at least partly, be
attributed to the larger variability of those variables during summer. In summer (JJA), the average
deviation of wind direction in observations at KPC_L is 40.3°. Whilst WRF is able to capture this
variability in wind direction (the average deviation is 41.1°), there is sometimes an offset in the timing
of the wind direction change between WRF and observations. For example, after two weeks of
consistently northwesterly winds being observed at KPC_L between August 11 to 24, 2014, there was
a shift to northeasterly flow on the morning of August 25 2014 (Fig 5e). WRF successfully simulated
the long period of northwesterly winds, and the shift to winds from the northeast, however the change
in direction was simulated in the late evening of August 25 to early morning of August 26 (Fig. 5f),
leading to a bias of 156.9° on August 25. The northeasterly wind was only observed for 24 hours
before returning to westerly on August 26 (Fig. 5g). WRF was able to capture the short-lived timing
of the event, but 24 hours later. In this particular case, the wind direction error comes from the
boundary data, ERA-Interim. In ERA-Interim, the wind direction change starts on August 24 but
remains northerly until 18:00 UTC on August 25. It then remains northeasterly until August 27, which
is 24-hours longer than in near-surface observations. The later onset and more persistent flow from the
northeast in ERA-Interim likely led to the later onset of northeasterly flow in WRF.  Therefore, WRF
can capture both the predominant wind flow, and abrupt changes to the wind direction, along with
capturing even short-lived events, although the timing is occasionally shifted. Figure 5 also highlights
that whilst the annual mean bias for wind speed is less than 1.5 ms$^{-1}$ (Table 2), during certain periods,
WRF simulates higher wind speeds than observed. However, these are not unrealistic values for this
region, with a maximum observed wind speed of 20.2 ms$^{-1}$ and a maximum simulated wind speed of
22.3 ms$^{-1}$ for the KPCL location. The largest values and biases of wind speed occur during
particularly strong katabatic events (northwesterly wind direction during winter). This was also found
by Hines & Bromwich (2008) when using the same land surface scheme as in these simulations.

Overall, WRF performs well at simulating air temperature, humidity, downwelling radiation

and wind speed during the simulation period (Oct 2013 - Dec 2018). WRF struggles to as accurately
represent the wind direction, especially at KPC_L (which is likely due to the proximity of complex
topography to the KPC_L site), however the winds remain predominantly westerly to northwesterly,
which shows that WRF can capture the dominant katabatic process governing the wind directions.

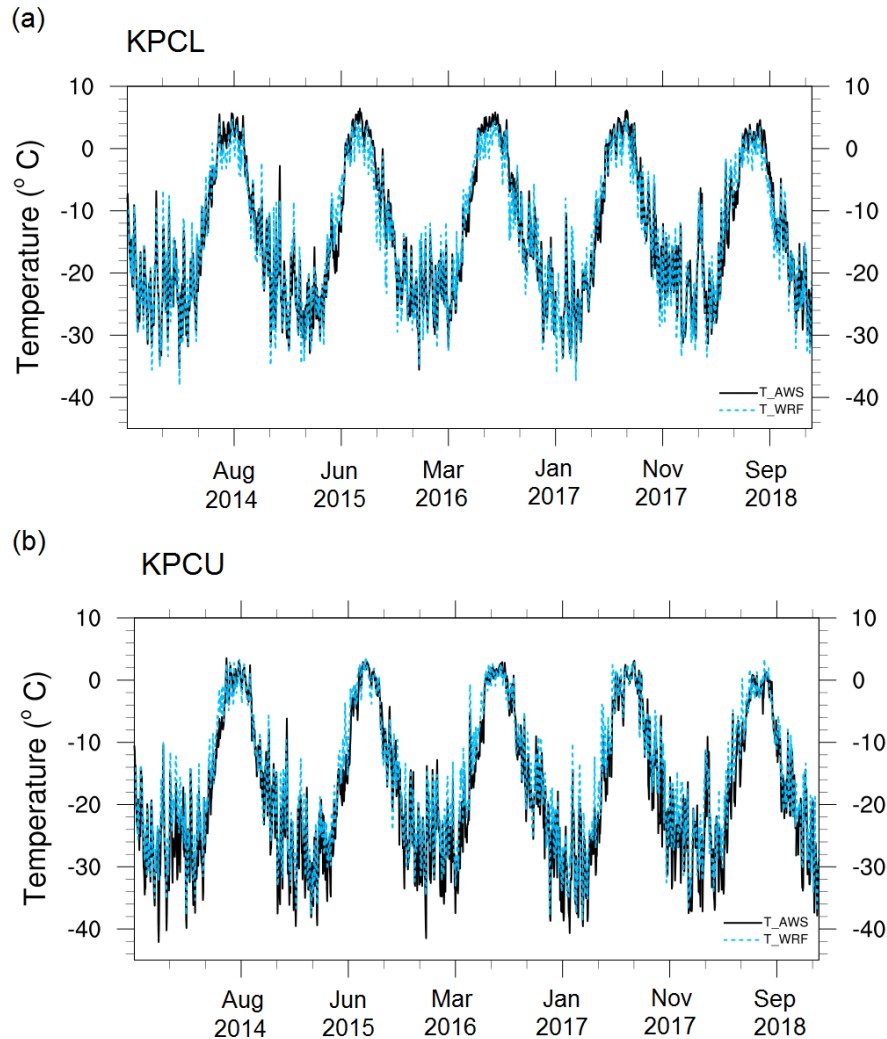


**Figure 3: The observed (black lines) and modelled (dashed blue lines) daily average air temperature at KPC_L (top) and KPC_U (bottom) from D03.**


**3.2 Model evaluation: Sub-daily Data**

To evaluate the ability of the model to simulate sub-daily values, the minimum and maximum daily near-surface values (from hourly output) are compared to observations, and the amplitude of the diurnal cycle of air temperature is also evaluated. Figure 6 presents the statistics for daily minimum and maximum air temperatures at the two locations in observations and WRF. The median values are well captured by WRF, especially for the maximum daily values, where a median value of -13.9°C is observed at KPC_U, and -14.0°C is simulated. Similarly, for maximum temperatures, the 75[th] quartile values are well captured by WRF (Fig. 6). For KPC_L, the minimum and maximum temperatures are colder in WRF than in observations. For example, the 25[th] percentile value for the minimum temperatures (far left bar in Fig. 6) is 3.8°C in observations, but 6.3°C in WRF. At KPC_U, the

opposite is true, where WRF simulates slightly higher temperatures than in observations. However,
overall, the range of minimum and maximum temperature values are well modelled by WRF.
The average daily maximum air temperature observed at KPC_L is -21.0°C in winter (DJF)
and increases to 3.0°C in summer (JJA). WRF simulates an average daily maximum of -20.9°C in
winter, which increases to 0.9°C in summer. The average daily minimum air temperature observed at
KPC_L is –25.9°C during winter and rises to 0.2°C in summer. WRF simulates an average daily
minimum air temperature of -26.5°C in winter and increasing to -2.3°C in summer. Therefore, WRF is
able to accurately simulate the winter minimum and maximum temperatures. WRF slightly
underestimates the air temperature during summer, however at KPC_U, this is within the error
estimate provided by the sensor manufacturer (Table 1), and for both locations the biases are not
statistically significant (Table 2).
Similarly, at KPC_U, the observed maximum temperature values are -24.1°C in winter and
0.1°C in summer. From WRF, the average maximum temperature is -22.5°C in winter and increases
to -0.1°C in summer. The observed minimum daily air temperature at KPC_U is -30.8°C during
winter and –3.5°C in summer. In comparison, in the WRF simulations, the average daily minimum
temperature is -27.4°C during winter and increases to -3.9°C in summer. WRF can therefore represent
the maximum and minimum daily air temperatures at KPC_U.
The annual-average observed diurnal air temperature amplitude is 5.6°C at KPC_U and 4.0°C
at KPC_L. The largest average diurnal cycle is observed during spring (MAM) at KPC_U (6.8°C) and
during winter at KPC_L (4.9°C). The WRF model simulated an average diurnal amplitude of 5.0°C at
KPC_U 4.7°C at KPC_L. The largest diurnal cycles are simulated during spring at KPC_U (6.2°C)
and during winter at KPC_L (5.5°C). Therefore, WRF accurately simulates the timing of the largest
diurnal amplitudes but overestimates the amplitude slightly at KPC_L, and underestimates it at
KPC_U, both by 0.6°C. The relatively large diurnal amplitude in winter may be counterintuitive given
that the glacier is located in the Arctic, where polar night (no solar radiation) prevails throughout
winter. However, the temperature variability is largest during winter over the glacier due to the more
frequent passing of storms across the Atlantic Ocean and the occurrence of 'warm-air events' from
easterly horizontal advection and increased longwave radiation from clouds (van As et al. 2009,
Turton et al. 2019a). Warm-air events are characterised by large (>10°C) temperature increases
between November and March, which can last for a number of days and, on average, occur 10 times
per year (standard deviation of 4.0) (Turton et al., 2019a). The variability can be further enhanced by
turbulent mixing from katabatic winds and the presence of föhn winds (Turton et al., 2019a).

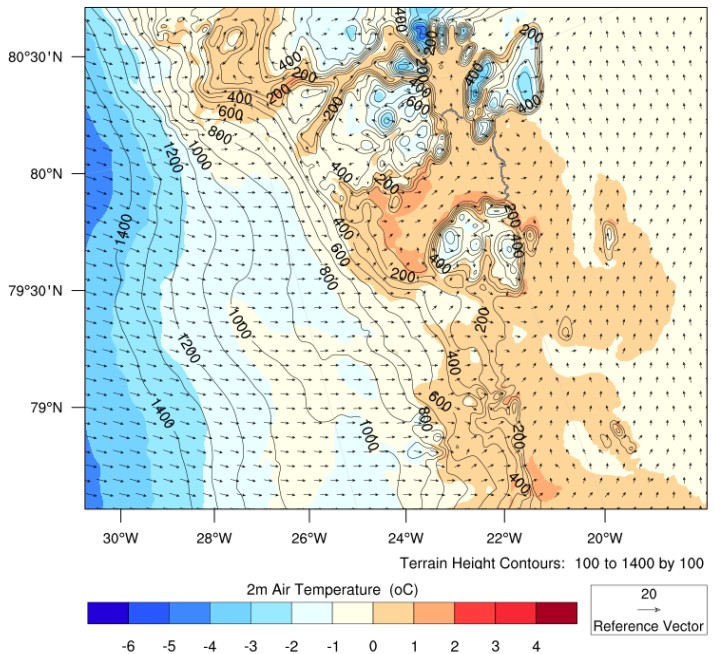

**Figure 4: The 2m air temperature (colours), wind vectors (arrows) and terrain height contours (black lines) for June 6 2015. The edge of 79°N glacier is shown by the dark grey line.**

The maximum hourly air temperature over the four years of data observed at KPC_L was on July 23, 2014 (8.1°C) (Fig. 6). WRF was able to replicate the processes responsible for the particularly warm day, as a daily maximum value of 4.5°C was modelled at KPC_U. At KPC_L, the maximum was simulated 24-hours earlier (6.5°C). The maximum values from WRF are slightly lower than observed (Fig. 6), but the timing of the maximum was accurate. The lower maximum values are likely linked to the negative mean bias in temperature simulated by WRF during the summer months (Table 2).

The absolute minimum hourly air temperature was observed at KPC_U on December 26, 2015 (-45.0°C) (Fig. 6) and on December 27, 2015 at KPCL (-37.2°C). Again, WRF was able to capture the events leading to the particularly cold December 2015 period. On December 27, the simulated minimum air temperature was -37.7°C at KPC_L and -37.8°C at KPC_U. The minimum daily values are warmer than those observed at KPC_U, but very similar to those observed at KPC_L. (Table 2).

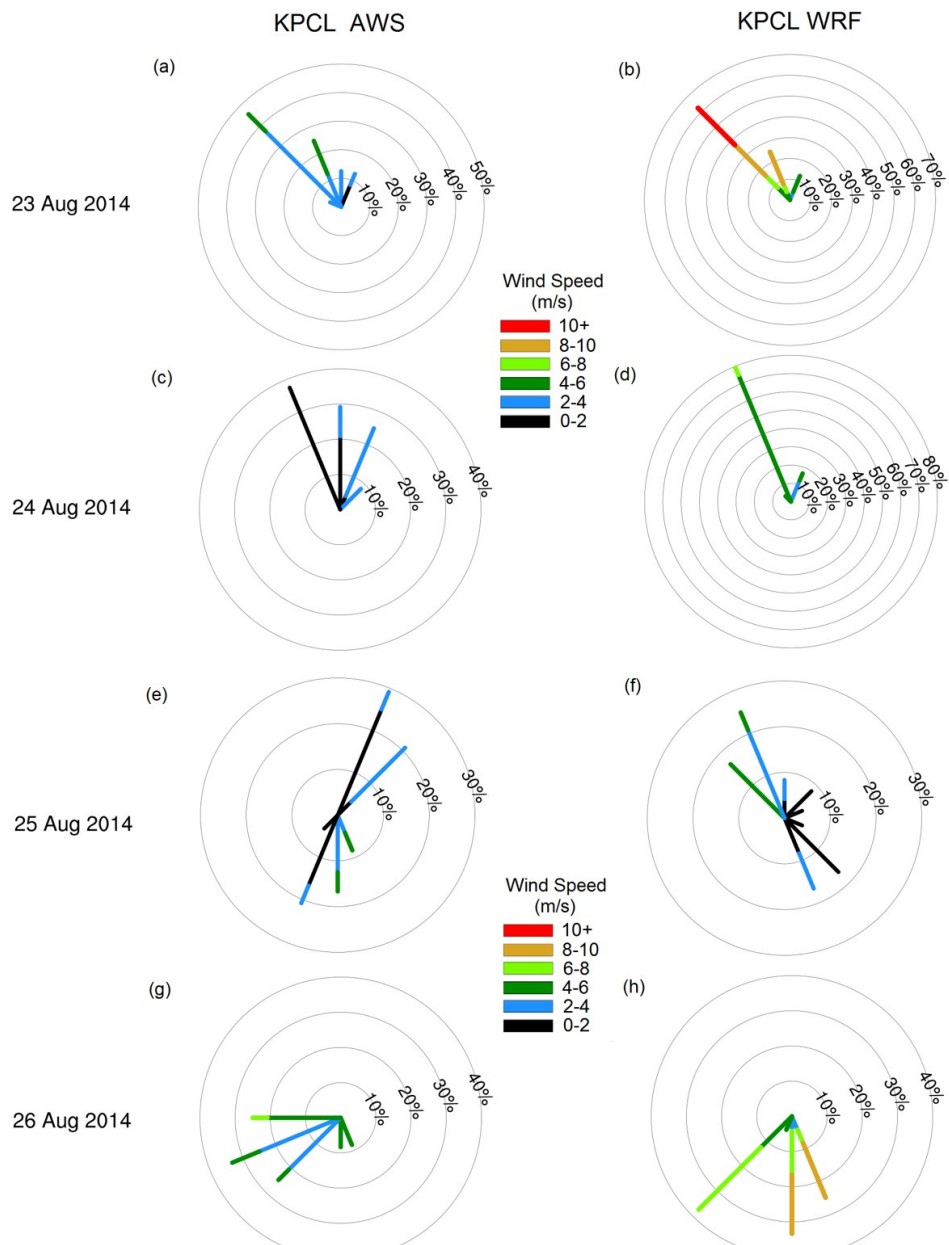

**Figure 5: Wind speed (colour) and direction (lines) for August 23 to 26, 2014, from observations (left panel) and WRF (right panel) at KPC_L location. The circles (and therefore length of the spikes) represent the frequency of the particular wind direction, with the percentage of occurrence written on the circles.**

## 4. Conclusions

Polar WRF has previously been extensively used in the Arctic (e.g Hines et al., 2011; Hines, & Bromwich, 2017; Wilson et al., 2011), including for Greenland (e.g DuVivier & Cassano., 2013; Turton et al., 2019a), for a number of applications. However, WRF runs have often been used for short case studies or performed at lower spatial resolution. This dataset provides high spatial and temporal resolution runs over multiple years (2014-2018) for an area of increased interest. Regardless

of the regular use of Polar WRF, it remains important to validate the model for specific locations,
especially when downscaling to very high resolutions.

Overall, the mean biases are small and statistically insignificant between the Polar WRF runs

and the PROMICE observations at both the lower and upper stations near 79°N glacier. The $R^2$
values are high for air temperature, humidity and wind speed, but less so for wind direction at
KPC_L. The wind direction is more variable in summer than in other months, and whilst WRF is able
to simulate the increased variability, large biases can arise due to inconsistent timing of wind direction
changes between WRF and observations over short periods of 24-hours or less. However, as WRF is
able to replicate the short-lived events and the predominant northwesterly winds of katabatic origin,
we can conclude that the NEGIS_WRF can be used for further studies of the near-surface meteorology
of the 79°N glacier. This dataset will be useful for many other applications in a number of fields
including the atmospheric and cryospheric sciences, and as input to hydrological, ice sheet and ocean
models, subject to appropriate validation.

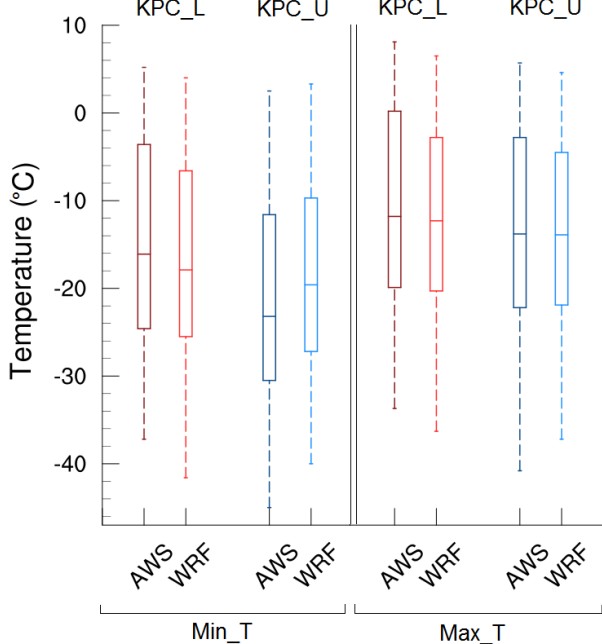



**Figure 6: Box plot representing the minimum (left) and maximum (right) daily temperature**
**values at KPC_L (red) and KPC_U (blue) locations, from both observations (darker colours)**
**and WRF (lighter colours).**

**5. Data Availability**
The atmospheric dataset, NEGIS_WRF resolves for the first time, the meteorological conditions over
the northeast region of Greenland (5km) and 79°N glacier region at the kilometre scale over a period
of five years (2014-2018). More than 50 variables are available (near-surface and on 60 atmospheric
levels) at up to hourly temporal resolution (for the 1 km domain), including meteorological and
radiative fields. Daily mean values for near-surface temperature (2m), specific humidity (2m), skin
temperature, and U and V wind components (10m) are available online (Turton et al 2019b:
doi.org/10.17605/OSF.IO/53E6Z) for the 1km and 5km domains from 2014-2018. As the output
frequency from D01 (25km resolution) was once per day, the available values are instantaneous daily
values at 00 UTC, as opposed to daily means. Furthermore, 4-D variables of temperature, humidity, U
and V wind components, geopotential and pressure are available on model levels at the same
frequency as the near-surface variables. For other variables, or more frequent output, please contact
the lead author, and these can be made available. Due to the large amount of data, these are not stored
online, but at the Regional Computation Centre Erlangen (RRZE) in Germany.
**6. Author Contributions**
JVT wrote the paper, ran the WRF model and evaluated it against the observations. TM and EC
contributed to the research concept, discussion, optimisation of the simulations and manuscript
refinement.

**7. Competing Interests**
The authors have no competing interests.
**8. Acknowledgements**
We thank Dirk van As from GEUS for his assistance with the PROMICE data and to Keith Hines for
the Polar WRF code. The authors also thank two anonymous reviewers and Dr Yasuhiro Murayama
for improving and editing our manuscript. This work was supported by the German Federal Ministry
for Education and Research (BMBF) and forms part of the GROCE project (Greenland Ice
Sheet/Ocean Interaction) (Grant 03F0778F). We acknowledge the High Performance Computing
Centre (HPC) at the University of Erlangen-Nürnberg's Regional Computation Centre (RRZE), for
their support and resources whilst running the Polar WRF simulations.

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
