# Peer review of "High-resolution (1 km) Polar WRF output for 79°N Glacier and the Northeast of Greenland from 2014-2018 3 Jenny V. Turton1, Thomas Mölg1, Emily Collier1 4 5 1Climate System Research Group, Institute of Geography, Friedrich-Alexander University, Erlangen"

_Earth System Science Data, 2019_

## Referee Comment (RC1) · Anonymous Referee #1 · 27 Nov 2019

General comments:

In this paper, the authors introduce numerical atmospheric model simulation data using the Polar Weather Research and Forecasting (Polar WRF) model applied in the Greenland. The authors set three model simulation domains: D1 covering the entire Greenland (horizontal resolution is 25 km), D2 covering the northeast Greenland (horizontal resolution is 5 km), and D3 covering the North East Greenland Ice Stream (NEGIS), where ice sheet thinning, retreat, and surface melt are accelerated recently, (horizontal resolution is 1km). First, they did dynamical-downscaling of the ECMWF (European Centre for Medium range Weather Forecast) ERA-Interim atmospheric re-

analysis dataset using Polar WRF in the D1 domain, then, did additional two dynamical downscaling calculations using Polar WRF in the D2 and D3 domains accordingly. In this paper, the performance of the model simulations for the D3 setting are presented and discussed in terms of 2 m air temperature and specific humidity, and 10 m wind speed and direction by comparing them with the PROMICE in-situ atmospheric measurement data. The authors argue very high-resolution model simulation data from the D3 configuration are very valuable because the dataset can be used for a wide variety of applications ranging from atmospheric dynamics studies, to input for hydrological and oceanic modelling studies. However, at present, this reviewer has the following concerns:

1. The open-source numerical simulation model (Polar WRF) itself is not developed by the authors. In my opinion, it implies that anyone who has enough scientific budgets can do this kind of numerical simulations relatively easily, so that the data lack high-level originality.

2. The authors do not validate the D3 model simulation results in terms of downward shortwave and longwave radiations, as well as surface (snow and ice) mass balance. Because downward radiations and precipitation are very important input parameters for hydrological and oceanic models, their argument "the dataset can be used to input for hydrological and oceanic modelling studies" is not supported by any objective evidence. Please note almost no direct precipitation measurement data on the Greenland ice sheet are available, so, usually, polar regional climate models are validated in terms of surface mass balance to confirm the models' performance simulating precipitation.

In the following part, this reviewer gives specific comments. Please note that page and line numbers are denoted by "P" and "L", respectively.

––––––––––––––––––––––––––––––––––––––––––––––––––––––––––––––––––––––––––

Specific comments (major)

P. 1, L. 17 ∼ 19: In order to argue "The dataset, (Turton et al, 2019b: doi.org/10.17605/OSF.IO/53E6Z), is now available for a wide variety of applications ranging from atmospheric dynamics, to input for hydrological and oceanic modelling studies", the authors should show model validation results in terms of downward short-wave and longwave radiations as well as surface mass balance.

P. 2, L. 44: However, recently, there are several attempts that applying high-resolution non-hydrostatic polar regional climate models in Greenland (Mottram et al., 2017; Niwano et al., 2018)

P. 2, L. 46: I think this model simulation by the authors is not "novel", because the model itself is not developed by the authors, which implies that anyone who has enough scientific budgets can do this kind of numerical simulations relatively easily.

Table 1: As far as I know, T2, Q2, WS10, and WD10 are not provided by GEUS. The provided T, Q, WS, and WD data are affected by surface height changes through accumulation/ablation.

Table 2: Please indicate coefficient of determination (R2) instead of correlation. For the model validation, indicating the R2 value is more general in my opinion.

Sect. 3.2: Why not directly showing comparison results for 1hour data?

———————————————————————————————————————————————

Specific comments (minor)

P. 1, L. 15: It is better to indicate time resolution here as well.

P. 2, L. 53: I think katabatic winds and warm-air advection can be simulated accurately even by a 5 km non-hydrostatic atmospheric model if the model considers detailed atmospheric and snow/ice physical processes in an appropriate manner.

P. 2, L. 54 ∼ 55: Which model configurations (D1 ∼ D3) can be used for this purpose? Please explain more.

[Figure]

P. 4, L. 83: What are the "other sources"? Please specify them.

P. 4, L. 109: Please explain why the Kain-Fritsch cumulus convection parameterization scheme was applied only for the D1 and D2 configurations.

P. 5, L. 121∼ 122: For snowpack, how deep do the authors consider in the model? Also, how did the authors confirm "the snowpack was adequately spun up before the onset of the accumulation season"?

––––––––––––––––––––––––––––––––––––––––––––––––––––––––––––––––––––––––

Technical corrections

Please unify notations of the model domains throughout the manuscript. At present, there are three types of notations like "d02", "D02", and "D2".

Figure 1: Please indicate the NEGIS area in this map as well.

––––––––––––––––––––––––––––––––––––––––––––––––––––––––––––––––––––––––

References

Mottram, R., Nielsen, K.P., Gleeson, E., Yang, X.: Modelling Glaciers in the HARMONIE-AROME NWP model, Adv. Sci. Res., 14, 323–334, https://doi.org/10.5194/asr-14-323-2017, 2017.

Niwano, M., Aoki, T., Hashimoto, A., Matoba, S., Yamaguchi, S., Tanikawa, T., Fujita, K., Tsushima, A., Iizuka, Y., Shimada, R., and Hori, M.: NHM–SMAP: spatially and temporally high-resolution nonhydrostatic atmospheric model coupled with detailed snow process model for Greenland Ice Sheet, The Cryosphere, 12, 635–655, https://doi.org/10.5194/tc-12-635-2018, 2018.

––––––––––––––––––––––––––––

---

## Referee Comment (RC2) · Anonymous Referee #2 · 20 Dec 2019

General comments:

This paper presents new high-resolution atmospheric modelling output for Northeast Greenland, in particular in the region of the 79°N Glacier, from 2014 to 2018. The authors are using the Polar Weather Research and Forecasting (Polar WRF) model and optimize it for their area of interest. This is an important contribution which most certainly will be useful for a broad range of scientists from different disciplines working, e.g., on the drivers of the observed thinning of the 80-km long floating glacier tongue during the past two decades. They are in need of high-resolution atmospheric data to study, e.g., surface melt at the outlet glaciers of the Northeast Greenland Ice Stream

(i.e., mainly the 79°N Glacier and Zachariæ Isstrøm), changes in the fast-ice cover (that opens up more regularly since 2000 (Sneed and Hamilton, 2016) potentially triggering increased calving), and/or ocean variability for a better understanding of melting along the bottom of the glacier tongue (e.g., wind stress driving warm Atlantic waters onto the continental shelf toward the glaciers (Münchow et al. 2019, accepted at Journal of Physical Oceanography)).

Data is accessible via the given identifier, of high-quality, and usable in its current format and size. The article is appropriate to support the publication of the data set. By reading the article and downloading the data set I would be able to understand and re-use the data set in the future.

Methods and materials are described in detail. Still, I am missing one example plot (e.g. temperature map with wind arrows on top) of a comparison between the 1-km resolution WRF output for the 79 North Glacier region and existing atmospheric models (e.g., RACMO2 at 11 km). This would be much more intuitive for the reader to assess the advantage of the new data set compared to existing ones as well as provide information on how good the coverage of the 80 km long and 20 km wide glacier tongue as well as the complex topography (e.g., Dijmphna Sound) is. Furthermore, I would appreciate a discussion on the given error estimates (see below for more details).

With respect to the high-potential of the data set being useful for a broad community of meteorologists, oceanographers, and glaciologists, the uniqueness of the data set providing 1-km high-resolution atmospheric data at the Northeast coast of Greenland (which certainly must have been cost-intensive to produce), and the completeness of the data set, I believe the paper should be published after some minor revisions outlined below.

Specific comments:

1. Structure of the manuscript: - The subsection 2.3 seems rather short and redundant. The model evaluation is what is shown and described in detail in the following result

sections. I would suggest to merge 2.2 and 2.3, i.e., describing briefly what you will use the observational data for, leading over to the result section. - You are lacking section 4. (Section 3 are the results and section 5 is the conclusion.) - To be more precise, I would suggest to change the subtitles of the result sections to sth like 3.1 Model evaluation: Daily-means 3.2 Model evaluation: Sub-daily data

2. References/citations: Some references were not appropriate/missing.

Line 22: Schaffer et al 2017 cited similar content in their introduction but this is not the content of their paper. Furthermore, mass loss of the Greenland ice sheet increased not only due increased ice discharge along the margin of the ice sheet (linked to the retreat of marine terminating glaciers) but also due to increased surface melt. Please elaborate this a bit more and refer to recent publications (e.g., Shepherd, A., Ivins, E., Rignot, E. et al. Mass balance of the Greenland Ice Sheet from 1992 to 2018. Nature (2019) doi:10.1038/s41586-019-1855-2).

Line 34: Is 1 m/yr given as an average melting over the entire glacier tongue? Thinning of the glacier tongue and it's variability in time has been also discussed in Mayer et al 2018. Furthermore, Wilson et al 2017 (The Cryosphere) and Mayer et al 2018 point out that thinning is mainly due to enhanced melt along the glacier base. Thus, surface melt (triggered by atmospheric changes) seems to be of minor relevance. However, the atmosphere may be relevant e.g. for driving oceanic heat toward the glacier (Mün-chow et al 2018, accepted at Journal of Physical Oceanography) and below the glacier tongue. If there is space, you could include these information to point out the relevance of a better understanding of atmospheric conditions to study the observed changes at the 79°N Glacier.

Lines 36-39: I would suggest to compare/list atmospheric modelling studies only or make more clear what kind of models were used in the listed publications. Schaffer et al 2017 do not use model data.

Line 89: "Analysis nudging" – I am not a modeler, so I am not fully sure how common it

is to use analysis nudging. I suggest to give a reference or add a very brief explanation on how this works.

3. Scientific questions/add-ons

Line 43-45: I suggest to point out why a 1-km resolution makes a difference in the coastal area of Greenland. One reason that you did not mention (or I missed it) is, that the topography along the coast is very steep and complex with a number of narrow fjords and small islands most likely channeling/blocking/steering the wind in your area of interest.

Line 88: Did one of the studies show that SST and sea ice concentration from the AVHRR compare well to other observations/satellite data?

Lines 93-97: A map showing Spalte Glacier and marking the open-water grid points would certainly help to better understand what you are describing. Are you also referring to the fast-ice cover named Norske Øer Ice Barrier (Sneed and Hamilton, 2016, Annals of Glaciology) here? Furthermore, I would split this long sentence into two.

Lines 97-98: "given the small area of calving at 79°N during this period." – I understand what you like to say but I think you should be more precise. You are talking about a (negligible) area change caused from calving at/advancing of the glacier front. Furthermore, it is not clear to me which time period you are referring to. Please specify.

Figure 2: Please add the shape (in white?) of the 79 North Glacier and point out its location. If possible, also add the location of Spalte Glacier and the approximate extent of the fast-ice cover. I believe that the dark blue color is missing in the Figure. At least I cannot distinguish areas deeper sea level from areas between 0 – 200 m height. Just from the color code, it looks like the islands along the coast are half under water.

Section 2.3: As stated above you may want to merge/skip this section. In case you keep it, I like to make you aware that it reads as if you compare air temperatures for model evaluation only, which is not the case.

Table 2, Line 163: I am not fully sure what the correlation coefficient refers to. Do you correlate the time series from WRF and AWS data over the entire measurement period? What do you mean by "annual correlation"? Do you correlate the annual means, i.e., 5 time steps only?

Line 174: "is more variable" – better use sth like "deviates more from the ASW data". Is that maybe due to the very steep topography presumably not covered by your 1-km resolution? Please discuss throughout the whole manuscript what may cause the described errors. How big are errors in the wind direction measured at the automatic weather stations?

Line 178: "is simulated better" – better use "more accurate" or "the model performs better in simulating..." or give specific numbers

Line 197: "WRF can simulate much higher wind speeds than observed" – Are these higher wind speeds (more) realistic? I am missing an interpretation/assessment/discussion of this result.

Line 199: "WRF struggles to as accurately represent the wind direction" – Please see my answer above to Line 174. I could imagine that this is a common problem at places with very steep and rugged topography along the Greenland coast, is it? How accurate is wind data measured at weather stations?

Line 224: Any idea why WRF underestimates summer air temperatures?

Line 236-239: On which time scales (how many days) do warm-air events occur? Can they really explain larger diurnal variability?

Technical corrections:

Lines 56: Repetition of the word "fields" in one line. Please rephrase. Furthermore, I would rephrase the sentence to "Here we present an evaluation ... to demonstrate the applicability ..."

Line 93: "floating tongue" – do you refer to the 79 North Glacier only? Otherwise it should be plural.

Line 103: spacing between both sentences is missing

Lines 102 – 112. This sentence is very long. I suggest to use bullet points for listing the different parameterizations.

Lines 155-: I suggest to shift the Table 2 to a new page, i.e., there should be some text directly after the heading of section 3.1.

Table 1: Please add °N/°W for units of the Location.

Figure 3: I suggest to use the same limits for the y-axis in a and b for easier comparison. (It would be easier to see that it gets colder at KPCU in winter.)

Figure 4: Please give more details in the Figure captions. What do the percentages tell us? Why are maximum ranges of percentages different?

---

## Author Comment (AC1) · 16 Jan 2020

Dear Editor and reviewers,

We would like to thank the two anonymous reviewers and yourself for the feedback on our manuscript. We have addressed point-by-point both the major and minor points raised by the reviewers below and in an attached PDF with different colours and font for our response. We have addressed all comments and suggestions, and have implemented the vast majority of them in the updated manuscript. We believe this has improved the manuscript quality. We hope that you find our response satisfactory and see that our dataset and manuscript as a good fit for the ESSD journal.

[Figure]

Kind regards, Jenny Turton, Thomas Mölg and Emily Collier.

Response to Reviewer 1

1: The open-source numerical simulation model (Polar WRF) itself is not developed by the authors. In my opinion, it implies that anyone who has enough scientific budgets can do this kind of numerical simulations relatively easily, so that the data lack high level originality. Whilst we agree with the reviewer that we do not develop the model, and also that, due to its open source and well documented nature many people can use WRF, we do not think that this detracts from the novelty of the manuscript. If we were to develop the model, or aspects of it, the results would likely be published in a model development journal, as opposed to a data journal. This dataset meets the aims of the journal in that high-quality data can be reused at a benefit to the earth system science community. Furthermore, we have conducted sensitivity studies and worked with the WRF model for over a decade, which means we are able to simulate this complex region using the most appropriate physics options (of which there are hundreds of combinations and many which would not be applicable or justifiable in the polar regions) as opposed to the default values which have unfortunately been used in many papers incorrectly. Furthermore, conducting these model runs at such a high spatial and temporal resolution took a large amount of computer time and money, which many colleagues in the scientific community do not have direct access to. Published papers within the ESSD journal include the use of ERA5 or other reanalysis products which are available online, and output from other models which were not developed by the authors, therefore we believe our paper to be within scope of this journal.

2: The authors do not validate the D3 model simulation results in terms of downward shortwave and longwave radiations, as well as surface (snow and ice) mass balance. Because downward radiations and precipitation are very important input parameters for hydrological and oceanic models, their argument "the dataset can be used to input for hydrological and oceanic modelling studies" is not supported by any objective evidence. Please note almost no direct precipitation measurement data on the Greenland ice

sheet are available, so, usually, polar regional climate models are validated in terms of surface mass balance to confirm the models' performance simulating precipitation. Thank you for this comment. We have now included a validation of the downward shortwave and longwave radiations as suggested, in table 2 and in section 3.1 of the results. We refrain from including any mass balance analysis or validation, as there is an ongoing project which is using the atmospheric WRF data as input to a mass balance model and will include a comparison to the WRF data. Please also see our response to P5 L 121-122 (about the snowpack), as it is relevant here. Furthermore, we have added the following "subject to appropriate validation" to the manuscript when referencing the datasets potential uses.

Major Comments

P. 1, L. 17 âĹij 19: In order to argue "The dataset, (Turton et al, 2019b: doi.org/10.17605/OSF.IO/53E6Z), is now available for a wide variety of applications ranging from atmospheric dynamics, to input for hydrological and oceanic modelling studies", the authors should show model validation results in terms of downward short-wave and longwave radiations as well as surface mass balance. Thank you for the comment. We have included a validation of the results for downward shortwave and longwave radiation and agree that it is useful for future users.

P. 2, L. 44: However, recently, there are several attempts that applying high-resolution non-hydrostatic polar regional climate models in Greenland (Mottram et al., 2017; Niwano et al., 2018). Thank you for highlighting this missing information. We have now included the following sentence: "Recently, there have been attempts at modelling the polar regions using non-hydrostatic regional climate models, including HARMONIE-AROME at 2 km resolution for the Southwest of Greenland (Mottram et al 2017b), and the NHM-SMAP at 5 km resolution for the whole of Greenland (Niwano et al 2018). However, the Mottram et al (2017b) study does not include the northeast of Greenland, and the focus of the Niwano et al (2018) study was to improve the surface mass balance estimates, as opposed to providing output for a more general atmospheric

sense".

P. 2, L. 46: I think this model simulation by the authors is not "novel", because the model itself is not developed by the authors, which implies that anyone who has enough scientific budgets can do this kind of numerical simulations relatively easily. Please see our response to the major questions above. However, we have removed the word novel here, as it does not alter the sentence, nor the impact of the paper.

Table 1: As far as I know, T2, Q2, WS10, and WD10 are not provided by GEUS. The provided T, Q, WS, and WD data are affected by surface height changes through accumulation/ablation. Yes, you are right. We have now altered table 1 and the citation to reflect this. Furthermore, we have added in a number of sentences in section 2.2 to explain our convention. This section now reads as: Observations are not measured at exactly 2m above the surface but vary with accumulation and ablation. Over bare ice, the sensor is 2.6m above the surface (van As et al. 2011). To clarify that the observations represent near-surface conditions, and are compared with 2m and 10m model output, we use the abbreviation T2m to represent near surface temperature, and WD10 for wind direction at 10m for both modelled and observed variables.

Table 2: Please indicate coefficient of determination (R2) instead of correlation. For the model validation, indicating the R2 value is more general in my opinion. Thank you for your comment. This has now been changed in Table 2 and throughout the manuscript.

Sect. 3.2: Why not directly showing comparison results for 1hour data? We are only comparing the WRF data with observations for the purpose of highlighting its skill and weaknesses for future users. We advise that any future users validate the model output for their own purposes and in a more detailed manner, therefore we chose only to show daily averages. To provide some information about the sub-daily skill, we discuss the diurnal temperature cycle and min/max values only.

Minor comments

P. 1, L. 15: It is better to indicate time resolution here as well. Included. The section now reads: "Here we present a high spatial- (1 km) and temporal- (hourly for the inner domain) resolution atmospheric modelling dataset."

P. 2, L. 53: I think katabatic winds and warm-air advection can be simulated accurately even by a 5 km non-hydrostatic atmospheric model if the model considers detailed atmospheric and snow/ice physical processes in an appropriate manner. Yes, we agree, as the warm-air advection processes can be seen in our 5km output too, but of course with less detail around the complex topography. We have altered the sentence to highlight that less than 5km would be good for these processes, and to include suggestions from reviewer 2. It reads as: "With a horizontal resolution of less than 5km, many atmospheric processes are accurately resolved including katabatic winds and warm-air advection (Turton et al., 2019a). Furthermore, high-resolution output is crucial for the complex topography on the northeast coast, where steep and variable topography can channel or block the winds."

P. 2, L. 54 âĹij 55: Which model configurations (D1 âĹij D3) can be used for this purpose? Please explain more. We have included which domain will be used for each of these purposes.

P. 4, L. 83: What are the "other sources"? Please specify them. Compared to MERRA2 reanalysis specifically, which has now been included. In Turton et al 2019a, we compared ERA Interim and MERRA2 to observations and found that ERA Interim was the more accurate choice. We only looked at ERA Interim and MERRA2 based on a study by Reeves-Eyre and Zeng (2017), who found these two to be the most accurate for our study area. This section now reads as: This reanalysis product was more accurate at resolving mesoscale processes in the northeast of Greenland compared to MERRA2 reanalysis data and has previously been used to initialise WRF in this region (Turton et al., 2019a).

P. 4, L. 109: Please explain why the Kain-Fritsch cumulus convection parameterization

scheme was applied only for the D1 and D2 configurations. From the literature and experience of the mesoscale modelling community, resolutions as coarse as 4km can be sufficient to resolve convection explicitly in non-hydrostatic models but resolutions of 1km give the most confidence in results (Weisman et al 1997). Resolutions between 8 and 12km can resolve convective processes only partly and often with timing issues, so parameterisation is recommended (Weisman et al 1997). Depending on which literature is assessed, anywhere between 2 and 20km is the 'grey area', and there is contrasting evidence in regard to whether one should or shouldn't use a parameterisation in this resolution range. We opted to use it based on previous studies and sensitivity analysis which showed little difference in output if the parameterisation scheme was or wasn't applied. This sentence has been altered to better reflect this information: "Further parameterisations include: the Kain-Fritsch scheme for cumulus convection (Kain, 2004) (D01 and D02 only, as the 1 km resolution of D03 allows convection to be explicitly resolved). "

P. 5, L. 121âĹij 122: For snowpack, how deep do the authors consider in the model? Also, how did the authors confirm "the snowpack was adequately spun up before the onset of the accumulation season"? The model was spun up during September, when there is typically no snow left on the glacier and bare ice is visible (from observations), but any water on the glacier has started to freeze over. The snowpack is handled by the NOAH land surface model (Chen & Dudhia, 2001), with some improvements to better represent the polar regions in Polar WRF (Hines & Bromwich, 2008). Maximum snow depth at the location of the AWS stations is approx. 3m, in June 2018, a year that had a significantly larger snowfall than previous years, when looking at ERA5 data (Turton et al. in prep). We used the observations of snow depth further south (74°N) along the coast by Pedersen et al. (2016) to compare against NEGIS_WRF runs in our D02 (during testing), which includes this location. In 2014 (the only year which overlaps with Pedersen's study), they record a max snow depth of approx. 0.9m in June 2014. At this location from NEGIS_WRF, 0.8m was simulated. Typically, this location is drier than 79N due to the blocking topography. Below the snowpack, the land surface

model treats glacier areas as fully saturated and fully frozen soils (Chen & Dudhia, 2001., Hines & Bromwich, 2008). This is appropriate for NE Greenland due to the low temperatures at this location. However, this is a limitation of WRF in glacierised areas, and this is more of an issue in mid-latitude glacier settings (see Collier et al. 2015; doi:10.5194/tc-9-1617-2015 for details). It is well known that the glacierised processes in WRF are simplified (for example, densification is not included), which is another reason why the authors refrain from providing an evaluation of the WRF-simulated surface mass balance. Instead, a study is currently underway to force a SMB model with NEGIS_WRF atmospheric data for this region to provide a more rigorous estimate of the SMB. As you mention above, there are little to no precipitation observations for this region. Wang et al (2019 :doi:10.5194/tc-13-1661-2019) compared ERA5 products with observations of precipitation from buoys in the Arctic sea ice (and near to the NE coast) and found a good agreement between ERA5 and observations in terms of snowfall. We have included the following line to highlight these changes: "The model was initialised and the snowpack spun up in September before the onset of the accumulation season. The model produces similar magnitude snow depths to available observations (Pedersen et al. 2016). Due to limited snowfall and snow depth observations in this region, we compared cumulative snowfall to ERA5 products during testing, which have been shown to have a relatively good agreement with observations by Wang et al. (2019)."

Technical corrections

Please unify notations of the model domains throughout the manuscript. At present, there are three types of notations like "d02", "D02", and "D2". Thank you for pointing this out, we have now remained consistent with the D02 style of notation.

Figure 1: Please indicate the NEGIS area in this map as well. This has now been included based on the Khan et al. (2014) study.

Response to Reviewer 2

Specific comments:

Still, I am missing one example plot (e.g. temperature map with wind arrows on top) of a comparison between the 1-km resolution WRF output for the 79 North Glacier region and existing atmospheric models (e.g., RACMO2 at 11 km). Thank you for this comment. We have now included an additional image (now called Figure 4) of the temperature and wind over the region, to highlight its skill in the spatial sense. We have decided not to compare with another model, as this would mean that we include another dataset only to compare for one figure, and we have attempted to keep comparison to other data and interpretation minimal to keep within the scope of the journal. We attempted to find published figures from atmospheric models for this region, but were unable to find some for a good comparison. We think that including the new figure will provide readers with a sense of how much detail is provided in 1km resolution runs.

1. Structure of the manuscript: - The subsection 2.3 seems rather short and redundant. The model evaluation is what is shown and described in detail in the following result sections. I would suggest to merge 2.2 and 2.3, i.e., describing briefly what you will use the observational data for, leading over to the result section. - You are lacking section 4. (Section 3 are the results and section 5 is the conclusion.) - To be more precise, I would suggest to change the subtitles of the result sections to sth like 3.1 Model evaluation: Daily-means 3.2 Model evaluation: Sub-daily data 2. Thank you for your suggestion. We have removed subsection 2.3, by incorporating the relevant information into section 2.2 and including some introductory sentences into the relevant results section (3). We have also changed the subtitles as you suggested. The numbering was wrong, as 'data availability' was numbered as 4, but actually came after section 5 (conclusions). We have now numbered them correctly.

2. References/citations: Some references were not appropriate/missing.

Line 22: Schaffer et al 2017 cited similar content in their introduction but this is not the
content of their paper. Furthermore, mass loss of the Greenland ice sheet increased not only due increased ice discharge along the margin of the ice sheet (linked to the retreat of marine terminating glaciers) but also due to increased surface melt. Please elaborate this a bit more and refer to recent publications (e.g., Shepherd, A., Ivins, E., Rignot, E. et al. Mass balance of the Greenland Ice Sheet from 1992 to 2018. Nature (2019) doi:10.1038/s41586-019-1855-2). Thank you for this suggestion. We have changed the Schaffer et al 2017 citation to one more specific (Howat and Eddy 2011). We have also included more description of the mass changes and included relevant information from the Shepherd et al publication. We have also shuffled some of the sentences around after including this information. The section now reads as: "The large amount of mass lost from the Greenland Ice Sheet (GrIS) within the last few decades (approximately 3800 billion tonnes of ice between 1992 and 2018: Shepherd et al. 2019) has largely been located around the coast of Greenland, due to the thinning and retreat of marine-terminating glaciers (Howat and Eddy, 2011), and the surface mass loss in the ablation zone due to enhanced melting and runoff (Rignot, et al, 2015; van den Broeke et al., 2017). The surface mass balance of a glacier is largely controlled by regional climate through varying mass gains and losses in the ablation and accumulation zones, respectively. A recent study found that enhanced meltwater run off, connected to changing atmospheric conditions, was the largest contributor of mass loss for Greenland (52%) (Shepherd et al. 2019). The remaining 48% of mass loss (1.8 billion tonnes of ice) was due to enhanced glacier discharge, which has been increasing over time (Shepherd et al. 2019)."

Line 34: Is 1 m/yr given as an average melting over the entire glacier tongue? Thinning of the glacier tongue and it's variability in time has been also discussed in Mayer et al 2018. Furthermore, Wilson et al 2017 (The Cryosphere) and Mayer et al 2018 point out that thinning is mainly due to enhanced melt along the glacier base. Thus, surface melt (triggered by atmospheric changes) seems to be of minor relevance. However, the atmosphere may be relevant e.g. for driving oceanic heat toward the glacier (Münchow et al 2018, accepted at Journal of Physical Oceanography) and below the glacier tongue.

If there is space, you could include these information to point out the relevance of a better understanding of atmospheric conditions to study the observed changes at the 79âŮęN Glacier. Thank you for these comments. We have now altered the sentence to better describe the thinning and the relevance of the ocean and atmosphere on melting. We have included more literature that you suggest, which provides more detail and explanation of the glacier. This section now reads as: "However, 79°N glacier, with its 80 km long by 20 km wide floating tongue, has retreated by 2-3km between 2009 and 2012, and the surface of the tongue and part of the grounded section of the glacier are now thinning at a rate of 1 m yr-1 (Khan et al., 2014, Mayer et al. 2018). The glacier is at a crucial interface between a warming ocean and a changing atmosphere. The mass loss from the floating tongue is largely attributed to basal melting due to the presence of warm (1°C) ocean water in the cavity below the glacier (Wilson and Straneo 2015, Schaffer et al. 2017, Münchow et al. 2019). Even the grounded part of the glacier is characterised by large melt ponds and drainage systems (Hochreuther et al, in prep), suggesting atmospheric processes may also be at play. Furthermore, atmospheric processes may be responsible for driving the warm Atlantic water under the glacier tongue, which leads to melting of the glacier base (Münchow et al. 2019). 79°N glacier is of further interest because its southerly neighbour, Zachariae Istrom, recently lost its floating tongue (Mouginot et al., 2015)."

Lines 36-39: I would suggest to compare/list atmospheric modelling studies only or make more clear what kind of models were used in the listed publications. Schaffer et al 2017 do not use model data. Yes we agree that this would be wiser, we have therefore limited it to atmospheric modelling studies only.

Line 89: "Analysis nudging" – I am not a modeler, so I am not fully sure how common it is to use analysis nudging. I suggest to give a reference or add a very brief explanation on how this works. Thank you for this comment. We have now included a number of sentences and references to explain this briefly. This section now reads as: "Nudging is the process of constraining the interior of model domains towards the reanalysis data

which drive the simulation (Lo et al. 2008., Otte et al. 2012). Nudging has been found to improve simulations of the large-scale circulation (Bowden et al. 2012) and reduce errors in the mean and extreme values (Otte et al. 2012) from relatively long runs. We only nudge the outer domain (D01) to allow the higher-resolution domain to evolve freely."

3. Scientific questions/add-ons

Line 43-45: I suggest to point out why a 1-km resolution makes a difference in the coastal area of Greenland. One reason that you did not mention (or I missed it) is, that the topography along the coast is very steep and complex with a number of narrow fjords and small islands most likely channeling/blocking/steering the wind in your area of interest. Thank you. We have now included a sentence with this information. It reads as: "Furthermore, high-resolution output is crucial for the complex topography on the northeast coast, where steep and variable topography can channel or block the winds."

Line 88: Did one of the studies show that SST and sea ice concentration from the AVHRR compare well to other observations/satellite data? We have now included a little more description of the dataset and included a reference where readers can find out about validation of the data for different regions. This section now reads: "NOAA Optimum Interpolation 0.25° resolution daily data. This is a combination of data from the Advanced Very High Resolution Radiometer (AVHRR) infrared satellite and Advanced Microwave Scanning Radiometer (AMSR) (doi:10.5065/EMOT-1D34, data retrieved from https://rda.ucar.edu/datasets/ds277.7/, last accessed July 29 2019). In-situ ship and buoy data are used to correct satellite biases, leading to relatively low mean biases of 0.2-0.4K for SST data (more information on this dataset can be found in Banzon et al. 2016)."

Lines 93-97: A map showing Spalte Glacier and marking the open-water grid points would certainly help to better understand what you are describing. Are you also refer-ring to the fast-ice cover named Norske Øer Ice Barrier (Sneed and Hamilton, 2016,

Annals of Glaciology) here? Furthermore, I would split this long sentence into two. Thank you for these suggestions. The sentence has been split and now reads as: "Other small exposed water areas along the coast, which are permanently frozen except in July and August each year (Hochreuther, P., 2019 personal communication), were also changed to ice during all months except July and August."

We have changed Figure 2 based on suggestions from yourself and Reviewer 1. The revised figure shows the land-use properties of the inner domain, labels important regions and highlights the changed grid points. The sea ice concentration data is the only information provided about the sea ice conditions. The resolution of the data is $0.25°$, so will not include small polynyas, leads or ice breakup. The concentration of sea ice is much lower very close to 79N (0.1-0.4 fraction) and Zachariae than further north and east (0.8-1.0 fraction), which may represent the thin, fast ice of the ice barrier.

Lines 97-98: "given the small area of calving at 79âŮęN during this period." – I understand what you like to say but I think you should be more precise. You are talking about a (negligible) area change caused from calving at/advancing of the glacier front. Furthermore, it is not clear to me which time period you are referring to. Please specify. The time period is during our study period, from 2014-2018. The text has been altered to make that clear. There is little information about the amount of area lost to calving during this time, but from looking at the shape files of glacier extent and calving front locations available at cryoportal.enveo.at, it is clear that the area of change is negligible. This section now reads as: "The glacier extents are treated as static throughout the run, which is an appropriate approximation given the small and likely negligible area of calving of 79°N during our study period (see ENVEO, 2019 for calving front locations from 1990 to 2017)."

Figure 2: Please add the shape (in white?) of the 79 North Glacier and point out its location. If possible, also add the location of Spalte Glacier and the approximate extent of the fast-ice cover. I believe that the dark blue color is missing in the Figure. At least I cannot distinguish areas deeper sea level from areas between 0 – 200 m height. Just

from the color code, it looks like the islands along the coast are half under water. We have revised figure 2 to provide more information about the region in terms of land use and location names. We have included labels for 79N and Spalte glacier. The height contour information from the previous figure (now removed) has been included in Figure 4 (a new figure of the temperature and winds). Adding the shape of 79N is not fully possible, as there is no clear outline of the glacier other than its floating tongue. It joins the North East Greenland Ice Stream (NEGIS) along with 2 other glaciers, and therefore there is no clear outline of where 79N stops and NEGIS begins. We have chosen not to include the extent of the fast ice, as we do not discuss this in the paper, and it is not an additional data source in the model, but will only be represented by the sea ice concentration. Furthermore, this extent changes annually, and seasonally, which would make the plot quite complicated.

Section 2.3: As stated above you may want to merge/skip this section. In case you keep it, I like to make you aware that it reads as if you compare air temperatures for model evaluation only, which is not the case. Thank you for this. We have now removed section 2.3 and merged the relevant parts into 2.2, but removed the part which made it seem like we only compared temperatures.

4Table 2, Line 163: I am not fully sure what the correlation coefficient refers to. Do you correlate the time series from WRF and AWS data over the entire measurement period? What do you mean by "annual correlation"? Do you correlate the annual means, i.e., 5 time steps only? As suggested by Reviewer 1, we have now calculated the coefficient of determination ($R^2$) rather than correlation coefficient, so the values have changed in table 2 and throughout the manuscript. For these calculations, we use daily averages of hourly observations and model output from the entire study period from 2014-2018 for the 'ANN' values. This is what we meant by 'annual correlation'- data from throughout the years. As opposed to $R^2$ values calculated only for DJF and JJA months. We see how this is misleading, so have changed the sentence slightly.

Line 174: "is more variable" – better use sth like "deviates more from the ASW data".

Is that maybe due to the very steep topography presumably not covered by your 1-km resolution? Please discuss throughout the whole manuscript what may cause the described errors. How big are errors in the wind direction measured at the automatic weather stations? Thank you for these suggestions. This sentence now reads as: "The wind direction in WRF deviates more from the AWS data than for temperature and moisture, which is likely due to the particularly steep and complex topography of the region which may not be accurately represented by the model at 1-km resolution."

We have now included more information on the errors throughout the manuscript. Table 1 now has an added column with information on the error from the automatic weather station sensors. We have now included information and discussion around the errors throughout the manuscript, which will appear in red text, but specifically, a sentence has been added in section 3.1 which says: "Some of these [wind direction] errors may relate to measurement errors of the wind sensor, which is +/-3° (see Table 1)."

Line 178: "is simulated better" – better use "more accurate" or "the model performs better in simulating. . ." or give specific numbers Thank you. Changed to: "The model performs better at simulating the wind speed than the wind direction."

Line 197: "WRF can simulate much higher wind speeds than observed" – Are these higher wind speeds (more) realistic? I am missing an interpretation/assessment/discussion of this result. Thank you for this suggestion. We avoided interpreting the results in detail, as this is out of the scope of this journal. However, we have now included some references to other literature. Higher simulated wind speeds from WRF, when compared to ERA Interim data (as we do here) were also found for the south east coast of Greenland by Duvivier & Cassano (2015). Compared to ERA-Interim, these differences could arise due to the different treatment of snow- and ice-covered surfaces and sea ice between WRF and the model used in ERA Interim. These differences can be found in Duvivier & Cassano (2015). The higher wind speeds are not unrealistic, and we have explained this in the text with the following: "Figure 4 also highlights that whilst the annual mean bias for wind speed is less than

1.5 ms-1 (Table 2), during certain periods, WRF simulates higher wind speeds than observed. However, these are not unrealistic values for this region, with a maximum observed wind speed of 20.2 ms-1 and a maximum simulated wind speed of 22.3 ms-1 for the KPCL location. The largest values and biases of wind speed occur during particularly strong katabatic events (northwesterly wind direction during winter). This was also found by Hines & Bromwich (2008) when using the same land surface scheme as in these simulations."

Line 199: "WRF struggles to as accurately represent the wind direction" – Please see my answer above to Line 174. I could imagine that this is a common problem at places with very steep and rugged topography along the Greenland coast, is it? How accurate is wind data measured at weather stations? We have now included this information in Table 1, and included a discussion of the errors in this section. It is a common problem in rugged and complex terrain areas, especially with very local scale wind patterns and abrupt changes in wind direction as seen in the northeast of Greenland. Changes can be seen in red in the manuscript. Please also see response to Line 174 which also covers this topic.

Line 224: Any idea why WRF underestimates summer air temperatures? The mean bias in summer is largest at KPCL and is -1.8°C. At KPCU, it is within the error estimate from the sensor manufacturer (+/-0.2°C). The differences between the model output and observations are not statistically significant either. We have now included this in the text: "WRF slightly underestimates the air temperature during summer, however at KPCU, this is within the error estimate provided by the sensor manufacturer (Table 1), and for both locations the biases are not statistically significant (Table 2). "

Line 236-239: On which time scales (how many days) do warm-air events occur? Can they really explain larger diurnal variability? The warm-air events are relatively frequent (10 times per year on average) and can last for approx. 2-4 days. If you look at Figure 2 in Turton et al 2019 (doi: 10.1175/MWR-D-18-0366.1) you can see the impact of a number of warm-air events in a single winter season. In combination with the larger

variability in this season due to frequent storms and more variable weather conditions, they do have a clear impact on the winter variability.

We extended this section to make it clearer. It reads as: "However, the temperature variability is largest during winter over the glacier due to the more frequent passing of storms across the Atlantic Ocean and the occurrence of 'warm-air events' from easterly horizontal advection and increased longwave radiation from clouds (van As et al. 2009, Turton et al. 2019a). Warm-air events are characterised by large (>10°C) temperature increases between November and March, which can last for a number of days and, on average, occur 10 times per year (standard deviation of 4.0) (Turton et al. 2019a). The variability can be further enhanced by turbulent mixing from katabatic winds and the presence of föhn winds (Turton et al., 2019a)."

Technical corrections: Lines 56: Repetition of the word "fields" in one line. Please rephrase. Furthermore, I would rephrase the sentence to "Here we present an evaluation . . . to demonstrate the applicability . . ." Changed.

Line 93: "floating tongue" – do you refer to the 79 North Glacier only? Otherwise it should be plural. In this sentence, we are referring only to the floating tongue of Spalte Glacier, not for 79N also, so it remains singular.

Line 103: spacing between both sentences is missing Changed.

Lines 102 – 112. This sentence is very long. I suggest to use bullet points for listing the different parameterizations. Thank you, we have split this into two sentences now. The use of the semi-colons for the list is preferable in our opinion.

Lines 155-: I suggest to shift the Table 2 to a new page, i.e., there should be some text directly after the heading of section 3.1. Changed.

Table 1: Please add âŮęN/âŮęW for units of the Location. Included.

Figure 3: I suggest to use the same limits for the y-axis in a and b for easier comparison. (It would be easier to see that it gets colder at KPCU in winter.) Changed.

Figure 4: Please give more details in the Figure captions. What do the percentages tell us? Why are maximum ranges of percentages different? We have now included that the percentages and circles relate to frequency of a particular wind direction. The maximum ranges are different as the occurrence of a particular wind direction can be very frequent (80% of the time wind was NNW for 24 August in WRF) or relatively infrequent (25 Aug 2014 in observations had variable wind with a maximum of 30% frequency for NNE wind). Although we attempted to have the same percentages shown for the left and right panels, it was difficult to read and interpret the data if 80% was used for all of them. For the 25th and 26th of August we were able to keep the same scale for observations and WRF output. But where they differed considerably, we used more appropriate scales. The caption now reads as: "Figure 4: Wind speed (colour) and direction (lines) for August 23 to 26 2014 from observations (left panel) and WRF (right panel) at KPCL location. The circles (and therefore length of the spikes) represent the frequency of the particular wind direction, with the percentage of occurrence written on the circles."

Please also note the supplement to this comment:
https://www.earth-syst-sci-data-discuss.net/essd-2019-194/essd-2019-194-AC1-supplement.pdf
* * *

---

## Author Response (AR2)

Dear editor and reviewer, we appreciate the time taken to review the manuscript again, and also for the additional comments, which have clarified the manuscript. We have addressed all three of the minor comments and included the responses below with an updated manuscript (changes marked in red). We would be very grateful if the editor is able to make a decision based on the changes we have provided, as opposed to sending the manuscript for a further review. We ask this, as our changes are very minor and due to the long period between the previous submission (January 28) and reviewer feedback (April 10) due to the absence of the topical editor.

Please let me know if any further changes are required.
Best wishes
Jenny Turton, on behalf of all authors.

P. 3, L. 75: Could the authors define "convection permitting" here? It is because some readers (especially glaciologists) might not in detail about the meaning of this technical term.
Furthermore, the focus of the Niwano et al. (2018) study was to improve the surface mass balance estimates, as opposed to providing output for a more general atmospheric sense, and the model was not convection permitting. In convection-permitting models, typically for spatial resolutions higher than 5km, convection begins to be explicitly resolved. This can enhance the representation of convection and associated precipitation, as opposed to using a convection parameterisation scheme, (Pal et al., 2019).

P. 9, L. 267~269: Useful information. Could the authors explain more in detail about "another scheme" briefly here?
P. 9, L. L. 270: Related to the comment above, one half of "both schemes" is not clear in the current version of the manuscript.

[revised manuscript text omitted]